# Genome-Wide Analysis of the Auxin/Indoleacetic Acid (*Aux/IAA*) Gene Family in Autopolyploid Sugarcane (*Saccharum spontaneum*)

**DOI:** 10.3390/ijms25137473

**Published:** 2024-07-08

**Authors:** Xiaojin Huang, Munsif Ali Shad, Yazhou Shu, Sikun Nong, Xianlong Li, Songguo Wu, Juan Yang, Muhammad Junaid Rao, Muhammad Zeshan Aslam, Xiaoti Huang, Dige Huang, Lingqiang Wang

**Affiliations:** 1State Key Laboratory of Conservation and Utilization of Subtropical Agricultural Biological Resources, Guangxi University, Nanning 530004, Chinamjunaidrao@gxu.edu.cn (M.J.R.); zeshanaslam87@gxu.edu.cn (M.Z.A.);; 2Guangxi Key Laboratory of Sugarcane Biology, College of Agriculture, Guangxi University, Nanning 530004, China; 3National Experimental Plant Science Education Demonstration Center, College of Agriculture, Guangxi University, Nanning 530004, China

**Keywords:** sugarcane, *IAA* gene family, phylogenetic analysis, cold and salt stress, GFP-subcellular localization

## Abstract

The auxin/indoleacetic acid (*Aux/IAA*) family plays a central role in regulating gene expression during auxin signal transduction. Nonetheless, there is limited knowledge regarding this gene family in sugarcane. In this study, 92 members of the IAA family were identified in *Saccharum spontaneum*, distributed on 32 chromosomes, and classified into three clusters based on phylogeny and motif compositions. Segmental duplication and recombination events contributed largely to the expansion of this superfamily. Additionally, cis-acting elements in the promoters of *SsIAAs* involved in plant hormone regulation and stress responsiveness were predicted. Transcriptomics data revealed that most *SsIAA* expressions were significantly higher in stems and basal parts of leaves, and at nighttime, suggesting that these genes might be involved in sugar transport. QRT-PCR assays confirmed that cold and salt stress significantly induced four and five *SsIAAs*, respectively. GFP-subcellular localization showed that SsIAA23 and SsIAA12a were localized in the nucleus, consistent with the results of bioinformatics analysis. In conclusion, to a certain extent, the functional redundancy of family members caused by the expansion of the sugarcane IAA gene family is related to stress resistance and regeneration of sugarcane as a perennial crop. This study reveals the gene evolution and function of the *SsIAA* gene family in sugarcane, laying the foundation for further research on its mode of action.

## 1. Introduction

The plant hormone auxin, also known as indole-3-acetic acid (IAA), is crucial in promoting plant growth and development. It also regulates plant responses to environmental factors such as phototropism, gravitropism, thigmotropism, shade avoidance, and stress responses [1]. This physiological regulation is accomplished through changes in the expression of numerous genes responsive to auxin perception and signal transduction [2]. Auxin signaling is controlled by a repression/de-repression mechanism involving the transport inhibitor response 1/auxin signaling F-Box (TIR1/AFB), Auxin/indole-3-acetic acid (Aux/IAA), and auxin response factor (ARF) proteins within the TIR1/AFB pathway. As central repressors in this pathway, Aux/IAA proteins can interact with TIR1/AFB and ARFs, thus receiving significant attention in research. The expression of *Aux/IAA* genes is tightly regulated and can be influenced by various factors, including auxin levels, developmental stages, and environmental cues.

Typical Aux/IAA proteins comprise four highly conserved domains designated as I, II, III, and IV, which contribute to their functional properties as short-lived nuclear proteins [3]. Domain I contains a conserved leucine sequence (LxLxLx motif) that can recruit TOPLESS (TPL)/TPL-related (TPR) corepressors [4] and is responsible for the proteins’ repressive activity [5]. Domain II features a conserved degron, the GWPPV motif, that binds to the auxin receptor during signal transduction, leading to the ubiquitination and degradation of IAA factors, thus modulating the expression of downstream genes [4]. Domains III and IV include a carboxy-terminal PB1 (Phox and Bem1) domain that forms a dimer with the ARF protein PB1, inhibiting the expression of auxin-responsive genes [6]. The combined effects of these characteristic regions within the AUX/IAA and ARF family enable precise control over auxin signaling, orchestrating essential processes throughout plant development [7].

The Aux/IAA-ARF module is a key component of auxin signal transduction [7]. Within cells, members of the ARF family form ARF–IAA complexes by binding to IAA, which represses ARF’s transcriptional activity [8]. When plants reach specific developmental stages or encounter external stresses, IAA proteins can be degraded, leading to the dissociation of the ARF–IAA complex. Consequently, ARF initiates the expression of specific genes, triggering a series of growth and developmental responses [9]. Previous studies have confirmed that members of the Aux/IAA and ARF families significantly contribute to various developmental processes and stress responses across multiple plant species [7]. *Aux/IAA* genes are crucial in auxin-related plant growth, including embryogenesis and the development of various organs [8]. Furthermore, *Aux/IAA* genes participate in drought resistance, nodulation, and the facilitation of interactions between auxin and other hormones, including abscisic acid, cytokinin, and ethylene. For example, under drought conditions, members of the IAA protein family (*IAA5/6/19*) orchestrate a transcriptional cascade to maintain levels of aliphatic glucosinolates (GLSs) [10]. Understanding how these proteins function can provide valuable insights for developing stress-tolerant crops. In rice, *IAA* genes such as *OsIAA9* and *OsIAA20* are significantly upregulated under high salt conditions [11]. *OsIAA6* enhances drought resistance in rice by responding to drought stress [12]. Under both drought and salt stress, rice *osiaa20* mutant plants exhibit reduced proline and chlorophyll contents, increased malondialdehyde content, and elevated Na+/K+ ratios [13]. The ABA-responsive gene *OsRab21* is downregulated in *osiaa20* mutants and upregulated in *OsIAA20* overexpression lines, illustrating *OsIAA20’s* role in the plant’s response to drought and salt stress through the ABA signal transduction pathway [13].

Sugarcane is a model C4 crop, contributing about 80% of the world’s sugar and about 40% of ethanol production worldwide [14]. Recent advancements have led to the successful assembly of the genome sequence of haploid sugarcane *S.spontaneum* AP85-441 (1n = 4x = 32), enabling further exploration in sugarcane genetic research and molecular breeding [15,16]. Despite the agricultural importance of sugarcane, there is a lack of information on the comprehensive characterization and functional analysis of the Aux/IAA gene family in this plant. This study aimed to achieve the following objectives: (1) Systematic identification of sugarcane Aux/IAA genes. (2) Description of the conserved domains and cis-regulatory elements present in their sequences. (3) Exploration of the distribution of Aux/IAA genes in the sugarcane genome. (4) Analysis of the evolutionary relationships among these genes to understand their origins and divergence between different sub-groups. (5) Expression profiling of Aux/IAA genes in various sugarcane tissues and developmental stages, under different stresses. (6) Prediction of putative protein–protein interaction and confirmation of subcellular localization through GFP assays. This study offers valuable insights into the sequence characteristics, genomic distribution, evolutionary background, and functionality of Aux/IAA genes in sugarcane, utilizing advanced bioinformatics and genomic tools to enhance the understanding of the functional aspects of the sugarcane Aux/IAA gene family.

## 2. Results

### 2.1. Identification and Distribution of Aux/IAA Genes in Saccharum spontaneum Genome

In the sugarcane v20190103 genome, a total of 92 members of the *IAA* gene family were identified. The sugarcane *IAA* genes (*SsIAA1* to *SsIAA31*) were named based on homologous genes in rice. Alleles on homologous chromosomes A, B, C, and D were distinguished by the letters a, b, c, and d, respectively. Additionally, *SsIAA* genes that do not have homologous genes in rice were named *SsIAA32* to *SsIAA38*. It is important to note that while some members are clustered on the same branch, they may not be located on homologous chromosomes. In such cases, alleles are represented by numbers 1, 2, 3, and 4 (Table 1, Figure 1, Appendix A). Thus, the *SsIAA* names contain information about the orthologous genes in rice as well as the homologous genes within the sugarcane genome.

### 2.2. Phylogenetic and Chromosomal Distribution of IAA Genes in Saccharum spontaneum

Previous studies have revealed that phylogenetic analysis can help elucidate evolutionary relationships and predict the potential functions of various genes [17]. A phylogenetic tree was constructed using 123 proteins, including 92 sugarcane IAAs and 31 rice IAAs (Figure 1). Three distinct groups within the sugarcane *IAA* gene family were identified: Group I with 17 members, Group II with 41 members, and Group III with 34 members. Members of Group II exhibited a relatively lesser homology than members of the rice *IAA* family.

An uneven distribution of the 92 *SsIAA* genes across 32 chromosomes was observed (Figure 2). Except for Chr1D and Chr6A, which contained only one gene member, other chromosomes possessed multiple (2–7) gene members. Notably, Chr2C contained the highest number (seven) of *IAA* genes.

### 2.3. Motifs, Conserved Domains, and Gene Structure of the SsIAA Gene Family

Proteins’ conserved motif analysis revealed that the *IAA* gene family comprised a total of ten motifs (Figure 3 and Appendix A). Motif 1 was shared by almost all IAA members, while Motif 3 was possessed by all gene family members except SsIAA1 and SsIAA5.1d, highlighting the importance of these two motifs in maintaining normal protein structure and function. The distribution and the number of motifs varied among the IAA proteins. Within the same group, the motif composition was similar. Group I had the highest number of motifs (1–10). Interestingly, SsIAA36a had repetitions of motifs, including the 1st, 2nd, 3rd, 5th, 6th, 9th, and 10th, while Group II and Group III had similar motif compositions with varied numbers (2–5).

*SsIAA* family members consisted of a variable number of exons, ranging from 1 to 28 (Figure 3B). The gene structural analysis revealed that members within a group possessed a similar number of exons and gene structures. Group I members contained the highest number of exons followed by Group II while Group III genes possessed the lowest number of exons. Almost all *SsIAA* genes demonstrated domains encoded by multiple exons, except *SsIAA5.1a* and *SsIAA7p*, which had only one exon. The diversity of the gene structure might be attributed to evidence regarding the evolution of gene families and potential roles in various biological processes.

A multiple-sequence alignment was constructed using amino acid sequences of the 38 SsIAAs (Figure 4). Four conserved domains were identified (I, II, III, and IV). We found that 17 SsIAA family members shared all four conserved domains, while 25, 22, 32, and 35 proteins shared domains I, II, III, and IV, respectively. Eleven of the SsIAA family SsIAAs were found to contain nuclear localization signals (NLSs). The typical NLS, also called an SV40-type NLS, is located at the end of domain IV. The ββα motif (two β sheets and one α helices), which functions in the dimerization of Aux/IAAs, was also found within domain III and a majority of the SsIAAs.

Following MSA, a representative protein from each phylogenetic group three-dimensional protein structure of IAA domains was deduced from sequences employing homology-modeling approaches (Figure 5). The structural comparison indicated that at the start of domain IV, SsIAA15c (Group II) lacked β3 and β4 (Figure 5A), and SsIAA23 (Group III) possessed the canonical β3 and β4 (Figure 5B), while SsIAA18a (Group I) contained an extended α1 (Figure 5C). Except for these regions, other secondary structures exhibited structural conservation, suggesting the structural variations at the start of domain IV might have led to the functional diversity of these proteins, in conjunction with the additional domains.

### 2.4. Evolutionary and Collinearity Analysis of SsIAA Genes

The duplication and evolution of Aux/IAA genes in sugarcane were examined by employing gene models from two monocot species’ (rice and sorghum) and one dicot species’ (Arabidopsis) genomes. The sugarcane *IAA* genes are distributed across 32 chromosomes with intraspecific collinearity (Figure 6A). The analysis indicated that among nonhomologous chromosomes, chromosomes 3 and 7 exhibited the highest collinearity. Interestingly, there were no syntenic members of the *IAA* gene family on Chr5A, Chr6A, and Chr6C. During the genetic and phenotypic evolution, gene duplication was crucial to gene expansion and functional diversification [18,19]. Using collinearity analysis, we determined the number of gene duplication events for the *SsIAA* gene family in the *S. spontaneum* genome (Figure 6A, Appendix A). Our study identified 37 genes (40%) that originated through whole-genome Duplication (WGD) or segmental duplications, while 19, 2, and 3 genes evolved through dispersed, proximal, and tandem duplications, respectively. Contrarily, 5 genes were singleton, while 26 genes had unknown origin.

Inter-species genomic collinearity analysis identified 59 and 62 *SsIAAs* to be orthologs with sorghum and rice, respectively, while 6 *IAA* collinear pairs were identified between *Arabidopsis* and sugarcane (Figure 6B). As predicted, monocot species exhibited greater homology of *IAA* genes than *Arabidopsis*, which is related to genetic relationships and species evolution. Additionally, our *SsIAA* inter- and intra-species analysis revealed that chromosomes 3 and 7 showed the highest numbers and variety of syntenic genes.

### 2.5. Prediction of Cis-Acting Elements in the Promoters of Sugarcane IAA Gene Family

Cis elements in promoter regions play an essential role in controlling transcription and expression, which can deepen the understanding of the regulatory function of *SsIAA* genes [20]. The promoter sequence of the sugarcane *IAA* gene contains five types of cis-acting elements. These elements are related to plant hormone regulation, transcription, stress response, light response, and plant growth and development (Figure 7, Appendix A). Transcription activity-related elements such as enhancer regions (CAAT-box) (2043 count) and TATA-box (1931 counts) elements exhibited the highest abundance. For hormonal responses, MeJA-responsive, auxin-responsive elements (AREs), Gibberline-responsive elements (GARE), and abscisic acid-responsive elements (ABRE), respectively, exhibited 428, 104, 67, and 33 counts. These findings strongly indicate the involvement of *SsIAAs* in the early auxin regulation of sugarcane. For stress responsiveness, *SsIAA* promoters showed the highest presence of anaerobic induction elements (539), followed by anoxic specific inducibility (97), LTR (low-temperature responsiveness) (83), drought inducibility elements (64), and stress-responsive elements (15), suggesting potential roles for *SsIAAs* in response to stresses such as low temperature and drought. Among growth related cis elements, meristem expression exhibited the highest abundance (97), while light responsiveness (50), endosperm expression (14), and circadian control (10) elements were also present. This variety of cis elements indicates that the *SsIAA* genes might be involved in various biological activities, as links between various hormone reactions and other essential biological processes.

### 2.6. IAA Gene Family Expression in Sugarcane Growth and Development and Stress Responses

Plant growth and development, including tissue differentiation and response to abiotic stress, are regulated by auxin signal transduction, facilitated by *auxin/IAAs* [20]. These repressors respond to auxin and control downstream gene expression. Therefore, transcriptomics data of 38 *SsIAAs* were used for hierarchical clustering of expression patterns (Figure 8), primarily classified into three categories, leaf and stem tissues at seedling, pre, and mature growth stages; different leaf sections; and circadian rhythms. Based on tissue-specific expression, *SsIAAs* were grouped into four types (Figure 8A and Appendix A). The I, II, III, and IV clusters were specifically expressed in leaves, mature plants, seedlings, and pre-mature stems. Based on the tissue and developmental stage-specific expression, the *IAA* gene family may play an important role in the growth and development of stem tissues in sugarcane.

Similarly, four differentially expressed *SsIAA* gene clusters were observed for the leaf sections (Figure 8B and Appendix A). The I, II, III, and IV clusters of genes were differentially induced in distal leaves (leaf section 15), middle leaves (leaf sections 4, 5, and 6), leaf bases (leaf section 1), and basal leaf regions (leaf sections 2 and 3), respectively. Contrarily, three types of co-expressed clusters were observed for circadian rhythms (Figure 8C and Appendix A). The first group’s expressions were high at 10 p.m., 6 p.m., and 2 a.m.; similarly, the second group was upregulated from 8 p.m. to midnight, whereas the third group’s genes were upregulated in the early morning from 4 to 8 a.m.

Notably, *SsIAA17d*, *SsIAA23*, *SsIAA3b*, and *SsIAA30-2p* exhibited significant upregulation in stems compared to leaf tissues. For expressions among different leaf sections, *SsIAA17d*, *SsIAA14*, *SsIAA30-2p*, *SsIAA23, SsIAA29*, and *SsIAA15c* showed overall higher expressions compared to the rest of the *IAA* gene family members. Interestingly, the expressions of the genes, as mentioned earlier, were high in basal regions but declined in the distal regions in conformation to the short-lived nature of the IAA encoded proteins [6]. The expression of three genes of Group Ⅲ (*SsIAA15c*, *SsIAA33.2*, and *SsIAA29*) was relatively obvious compared to others. Similarly, the expressions of SsIAA17, SsIAA23, and SsIAA33.2 for circadian rhythms were significantly higher than other *SsIAAs*. In summary, based on three transcriptome datasets it could be determined that *SsIAA17d*, *SsIAA23*, *SsIAA3b*, *SsIAA30-2p*, *SsIAA14*, *SsIAA15c*, *SsIAA33.2*, and *SsIAA29* were important candidate genes for further functional characterization.

We examined the expression patterns of the nine *SsIAA* genes under cold and salt stress to uncover potential roles for the *IAA* gene family members in response to abiotic stresses (Figure 9). Under salt stress, the results indicated that except for *SsIAA19a*, which was downregulated, the other four genes (*SsIAA23*, *SsIAA7a*, *SsIAA18a*, and *SsIAA29*) exhibited positive inductions (Figure 9A). For the majority of genes, a 6 h salt-stress interval resulted in peaks of expression, while it started to decline at a 12 h interval. Similarly, exposure to cold stress led to a significant increase in the expression of *SsIAA13a*, *SsIAA23*, and *SsIAA9a* genes (Figure 9B). For cold stress, *SsIAA13a* and *SsIAA23* exhibited the highest expression at a 3 h interval, while the expression of *SsIAA9a* peaked at 6 h.

### 2.7. Subcellular Localization of SsIAA23 and SsIAA12.1a

The online tool WoLF PSORT was used to predict that most SsIAA proteins were most likely to be located in the nucleus (Table 1). From the preceding heatmap analysis of distinct tissues and stress treatments, it was evident that *SsIAA23* consistently displayed high expression levels. Hence, *SsIAA23* should be regarded as a noteworthy candidate hub gene with distinct functions of auxin-signaling transduction. To further characterize this gene, we performed GFP subcellular localization experiments of its protein along with SsIAA12.1a (Figure 10). The open reading frames of both genes (SsIAA23 and SsIAA12.1a) without stop codons were cloned into the pCAMBIA1300-35S-GFP vector. The *Nicotiana benthamiana* leaves were infiltrated with three pCAMBAI300-35S: SsIAA23-GFP and pCAMBAI300-35S: SsIAA12.1a-GFP constructs, and 60 h after infiltration, epidermal cells were seen under a confocal microscope. The microscopic images indicated that both proteins were localized to the nucleus, in conformation with the predicted locations (Figure 10).

### 2.8. The Protein–Protein Interaction Prediction of the IAA Gene Family

An interaction network comprising SsIAA proteins was established using the STRING website tools, based on their homology to proteins in rice, and it aimed to delve deeper into the potential connection of these proteins (Figure 11). To study the interaction between SsIAA family proteins, the regulatory network between SsIAAs and other proteins was constructed using rice as a reference. The prediction results showed 92 nodes in the interaction network and 86 SsIAA interactions with ARF proteins. Several of these genes functioned as hub genes within the regulatory network, which interacted with ARFs (Figure 11A). The identification of regulatory networks provides valuable information for better understanding the roles of IAA and ARF genes in development and stress responses. Most SsIAAs do not interact with each other; interestingly, SsIAA10 interacted with SsIAA19, SsIAA24, SsIAA14, and SsIAA12, which might suggest their co-regulation (Figure 11B).

## 3. Discussions

Auxin is a major plant signaling transducer that regulates growth and development [21]. Aux/IAAs bind to ARFs, which are also implicated in auxin gene expression responses, and suppress the expression of downstream genes [22]. However, there is not much information on *Aux*/*IAA* genes in sugarcane yet. To shed light on the potential role of *Aux/IAAs* in sugarcane plants’ growth and development and responses to stress, we identified and analyzed all of the *Aux/IAA* genes in sugarcane using the entire genome of autopolyploid cultivated sugarcane in this study. We also used qRT-PCR to examine the expression pattern of nine *SsIAA* genes during cold and drought stress.

With the recent advancements in whole-genome sequencing tools, the *Aux/IAA* gene family members have been found at the entire genome levels of many crops and other plant species. There were notable differences in the number of *Aux/IAA* genes in different species; for example, there was just 1 in *Marchantia polymorpha* [23], 41 in alfalfa [24], 28 in Arabidopsis [10], 18 in papaya [25], 89 in turnip [26], 119 in *Brassica napus* [2], 63 in soybean [27], and 19 in *Prunus mume* [28].

We identified 92 *SsIAA* genes in sugarcane using homology-based search methods (Appendix A). The number of genes identified suggests the second highest among the reported plant species. Due to sugarcane’s octoploid nature, 38 basic sets of genes were discovered, among which 31 genes were orthologous to rice, while 6 were unique to sugarcane. The whole genomic collection of 92 *SsIAA* genes was constituted by allelic and non-allelic compliments of 38 *SsIAA* genes on eight sets of four homologous chromosomes. Based on rice, with 31 *IAA* genes in rice plants of the grass family lineage, one would expect sugarcane to have ~124 *SsIAA* genes, instead of the current genomic collection of 92 genes. We speculate that during whole-genome duplication (WGD) of sugarcane, some alleles of *SsIAAs* were lost similar to the *Brassica napus IAAs* in dicots [2].

Analyzing the encoded proteins’ physicochemical features helped clarify the role of *SsIAA* genes. It was discovered that 92 SsIAA proteins exhibited a range of physicochemical traits (Table 1 and Appendix A). The number of amino acids of most of SsIAAs ranged from 139 (SsIAA26a) to 1904 (SsIAA36d), and the mean value of instability index was 54.3, which was higher than 40 (standard for comparison), indicating these are unstable proteins. The subcellular prediction analysis suggested that almost all SsIAA proteins are in the nucleus. These distinguishing characteristics of sugarcane Aux/IAA were comparable to those of Aux/IAA in most plants [29], which may be connected to Aux/IAA’s conservatism, suggesting that their roles are similar.

Using phylogenetic analysis, it is possible to clarify evolutionary links and provide predictions about the potential functions of genes [17]. Contrary to the previously reported five [30] and two clad classifications [31], the phylogenetic tree involving 92 SsIAA proteins and 31 rice orthologs exhibited partition in three large groups (Figure 1). We mapped all 92 *SsIAA* genes on 32 chromosomes in silico (Figure 2) and found various arrangements of these genes either in clusters or singly located. Following that, the homology and evolutionary origins of *Aux/IAA* genes were explored in a range of different species (Figure 6B). A total of four species were examined during this investigation. *Arabidopsis thaliana* was discovered to possess six syntenic *Aux/IAA* genes with sugarcane, suggesting that these genes may have been inherited from a common ancestor of earlier land plants. An earlier study in ginseng identified five syntenic *Aux/IAA* genes within *Arabidopsis* [32]. Conserved protein-motif and gene-structure analysis (Figure 3) revealed that among three phylogenetic clads, Group II possessed the highest number of motifs and exons, suggesting that this clad harbors genes and proteins of large sizes compared to the other two phylogenetic groups. The IAA protein domains are the characteristic domains of this gene family, and four small motifs or domains further constitute these domains. Therefore, we performed multiple-sequence alignments encompassing IAA domains among SsIAA members (Figure 4). The analysis revealed that only 17 proteins possessed all four domains, while only eleven possessed the C-terminal NLS domains. Furthermore, to understand protein structures we performed 3D protein modeling (Figure 5). The 3D homology-modeling structures revealed that only SsIAA23 (Group III) had secondary structures in canonical beta palates (β3 and β4). At the same time, SsIAA15c (Group II) and SsIAA18a (Group I) exhibited an unstructured loop and extended α1 helix of domain III, respectively. Taken together, the structural variations in the PB1 domain among different phylogenetic groups might be a factor in the auxin-signaling pathway’s varied activities, which, in turn, helps plants’ ability to respond to environmental changes through the various roles that *Aux/IAA* plays [33].

Correlation studies showed that promoters impact temporal and spatial variations in gene expression, and cis elements inside the promoter regulate gene function by interacting with trans-acting components [34]. Numerous promoter motifs linked to hormones and abiotic stress have been found in the *Aux/IAA* genes [35]. Drought inducibility (MBS), defense and stress responsiveness (TC-rich repeats), low-temperature responsiveness (LTR), and hormone-responsive elements (AuxRR-core, ABRE, TGA-element, and CGTCA-motif) were among the major cis elements identified by the analysis of the *SslAAs* promoters (Figure 7). *SssIAAs* may react to a range of stimuli, such as MeJA, auxins, GA, salicylic acid, ABA, drought, heat stress, salt, and low temperature, suggesting that these genes might be induced in stress response and/or phytohormone signaling. The AuxRE elements found in the promoters of several *SsIAA* genes interact with the downstream ARFs, which is crucial for the transcriptional regulation of the auxin pathway [36].

Many *SsIAA* genes exhibited specific upregulated expressions in pre-mature and mature stems compared to leaf tissues (Figure 8A and Appendix A), suggesting that these genes may control biological processes associated with stem development. In a previous study, 9 among the total 19 *PmlIAA* genes exhibited high expressions in the stem tissues of *Prunus mume*, suggesting their potential function in stem growth and development [28].

Additionally, *GmIAA45* and *GmIAA51* transcripts were found highly abundant in soybean shoots [27], while five *PeIAAs* (*PeIAA1*/*2*/*6*/*8*/ and *16*) were significantly expressed in stems of *moso bamboo* [37]. The *SsIAA* genes’ function in stem formation may offer prospective genetic materials for sugarcane breeding, as stems are the most commercially important parts of the plant [38]. Furthermore, Cluster I (*SsIAA17d*/*19a*/*23*/*14*/*4a*), in Figure 8A, was preferentially expressed in leaves rather than stems, suggesting that it might play roles in photosynthesis. Sugarcane is a representative C4 plant with an extraordinary light-usage capacity. One possible application of the grass-leaf development gradient model is the investigation of C4 photosynthesis and its regulatory elements [39]. *SsIAA* gene regulation of C4 photosynthesis was studied using the sugarcane leaf’s developmental gradient expression landscape. The C4 photosynthesis development pattern suggests that leaves steadily differentiate for active photosynthesis [40]. Interestingly, the genes that are highly expressed in stem tissue (Clusters II, III, and IV in Figure 8A) exhibited upregulation in the basal region of leaf sections (Figure 8B and Appendix A). Since sugarcane stems and basal leaf regions act as sinks [41], we speculate that the majority of *SsIAAs* might play roles in sugar transport and storage. Conversely, Cluster I genes (Figure 8A), which are specifically expressed in leaves, exhibited transcript abundance in the distal part of leaves (Cluster I, Figure 8C), which are active regions for photosynthesis. Based on expression analysis, we tentatively speculate that *SsIAAs* exhibit bifurcation in expression or function for active photosynthesis and sugar transport/storage. The transcriptome data for circadian rhythms revealed that most *IAAs* were upregulated from dusk to midnight (Figure 8C and Appendix A). Aux/IAA act as repressors, which bind and repress activator ARFs in the absence of auxin, blocking downstream target gene upregulation [42]. To enable ARF-mediated upregulation of auxin-responsive transcripts, including the Aux/IAA themselves, auxin stimulates the destabilization of the Aux/IAA protein. Auxin levels drop off at dusk, activating the negative feedback mechanisms to deactivate auxin-mediated signaling [36].

Throughout their life cycle, plants are regularly subjected to environmental stresses including desiccation, salinity, and cold, which impact their growth and development [43,44]. Numerous studies have demonstrated that the auxin-responsive genes are involved in various stress responses. Previous research revealed that salt and drought stress treatments caused a surge in poplar’s *Aux/IAA* gene transcripts [45]. Furthermore, tissue-specific genes exhibiting differential gene expression may play important roles in stress responses as well [46]. Therefore, we tested five *SsIAA* (*SsIAA19a SsIAA23, SsIAA7a, SsIAA18a,* and *SsIAA29*) expressions during salt stress using qRT-PCR. The results indicated that four genes’, including *SsIAA23*, expressions were upregulated under salt-stress conditions (Figure 9A). The positive induction of *SsIAAs* after salt stress was in conformation with the earlier reports in rice [11] and chickpeas, but opposite to the soybean [27], suggesting the trend may vary between different species. As a result of a mutation in *Aux/IAA14*, the auxin-signaling mutant solitary root 1 (*slr1*), when subjected to cold stress at 4 °C, exhibited an oversensitive reaction to the stress, suggesting the *IAA* gene’s role in cold tolerance [47]. Similarly, RNA-seq data in alfalfa [24], chickpea, and soybean [27] exhibited preferential upregulation under cold stress treatments. In conformation with the above studies, qRT-PCR experiments of four *SsIAA* genes under cold stress exhibited highly induced expressions (Figure 9B). Notably, *SsIAA23* was also upregulated in response to salt stress, suggesting this gene is involved in multiple stress responses.

Studying the subcellular distribution of proteins aids in the exploration of their biological roles [48]. Therefore, subcellular localization verification experiments were performed. The tested proteins SsIAA23 and SsIAA12a exhibited subcellular localizations in the nucleus (Figure 10) in agreement with their predicted locations (Table 1). In previous studies of ginseng [32] and *Dendrobium officinale* [4], Aux/IAA proteins also exhibited IAA-GFP signals in the nucleus, suggesting conserved subcellular locations of IAA proteins across species. Furthermore, in silico protein–protein interaction analysis predicted the interactions of IAA proteins with ARFs (Figure 11), which was in conformation with the reported Y2H interactions [4,32].

Recently, CRISPR cas9-mediated gene knockout studies have been reported to study the gene functions or to improve/increase sugarcane traits. For example, lignin contents in sugarcane have been reduced by knocking out the *Solim* transcription factor gene [49], while in another study herbicide resistance in sugarcane was improved through homology-dependent repair-mediated gene targeting of in the acetolactate synthase (*ALS*) [50]. The pursuit of identifying specific and novel candidate genes presents a promising approach to enhancing sugarcane’s stress tolerance and yield in this context. Therefore, novel candidate genes of the *Aux/IAA* gene family such as *SsIAA23*, identified and somewhat characterized in this research, warrant further functional studies using heterologous overexpression systems in yeast, *Arabidopsis*, tobacco, rice, or sugarcane (linked with the development of a transformation system in sugarcane) itself.

## 4. Materials and Methods

### 4.1. Identification of IAAs in Saccharum spontaneum

To discover *IAA* gene family members within the Saccharum spontaneum genome, complete protein sequences of IAA from *Oryza sativa* were acquired through the Rice Genome Annotation Project database (http://rice.uga.edu/index.shtml (accessed on 28 August 2023)). The latest genome version (v20190103) of sugarcane was downloaded from Monocots PLAZA 4.5 (https://bioinformatics.psb.ugent.be/plaza/versions/plaza_v4_5_monocots/organism/view/Saccharum%2Bspontaneum, accessed on 28 August 2023).

First, SsIAA protein sequences were identified using the local BlastP program in Bioedit software 7.2 [42] using *Oryza sativa* IAA proteins as query sequences with stringent thresholds of E value < 1 × 10^−5^, query cover > 50%, and protein identity > 30%. Duplicate sequences were removed from the search results, and putative member sequences were obtained. Subsequently, the conserved domains of candidate protein sequences were further identified using the NCBI “batch Web CD-Search Tool” (https://www.ncbi.nlm.nih.gov/Structure/bwrpsb/bwrpsb.cgi (accessed on 30 August 2023)) to detect each candidate protein as an IAA protein. Lastly, the online tools SMART (http://smart.embl-heidelberg.de/smart/set_mode.cgi (accessed on 5 September 2023)) and Pfam (http://pfam-legacy.xfam.org/ (Pfam ID: PF02309 accessed on 15 September 2023)) were used to conduct further verification of the presence of conserved domains in the search results, ultimately identifying the candidate SsIAA genes. Utilizing TBtools software (version 2.019) [51], the biochemical characteristics of each SsIAA protein, encompassing amino acid count, molecular weight, isoelectric point (pI), and instability index, were calculated.

### 4.2. Phylogenetic Analyses of SsIAAs

To explore the evolutionary relationship of SsIAAs, two phylogenetic trees were generated by utilizing all candidate protein sequences. The Muscle program with default parameters in MEGA-X [52] software was used for multi-sequence alignment analysis. Subsequently, phylogenetic trees were created using the neighbor-joining (NJ) approach with MEGA-X software (with 1000 replications for bootstrapping). To enhance visual presentation, the ultimate phylogenetic tree of the SsIAA families was elaborated and annotated using the online resource Interactive Tree of Life (iTOL) (https://itol.embl.de (accessed on 25 September 2023)).

### 4.3. Chromosomal Localization and Synteny Analysis of SsIAA Genes

The TBtools software (version 2.019) was utilized to extract the gene coordinates from the *S.spon* GFF3 files and draw chromosomal maps of *SsIAAs* on chromosomes. Based on this positional information, the chromosomal location image of SsIAAs was created. Based on the physical location information of genes on chromosomes, the *SsIAA* genes were renamed (*SsIAA1-SsIAA38*). To further analyze the intra-species synteny of SsIAA genes, the syntenic gene pairs of SsIAAs were identified through MCscanX analysis, and the synteny circos plot was visualized using the “Advanced Circos” program from TBtools software (version 2.019). Moreover, the “One step MCScanX” program in TBtools software (version 2.019) was utilized for the syntenic analysis of *IAA* genes in *Sorghum bicolor*, *Oryza sativa*, and *Arabidopsis*, and the results were visualized using the “Multiple Synteny Plot” program.

### 4.4. Gene Structure, Protein Conserved Motif, Domains, 3D Modeling, and Promoter Cis-Elements Analysis

The “Gene Structure View program” from TBtools v2.016 was employed to visualize the exon/intron structure of the SsIAA, using the genomic structure information (GFF) and gene ID data. The conserved motif configurations within the proteins encoded by SsIAAs were examined utilizing the online application Multiple Em for Motif Elicitation (MEME Version 5.5.3), accessible at (https://meme-suite.org/meme/) (accessed on 26 September 2023)). The parameter settings used were a maximum number of motifs set to 10 and a maximum width of 50. For domain analysis, 38 representative protein multiple-sequence alignments were subjected to ESPript 3 (https://espript.ibcp.fr/ESPript/cgi-bin/ESPript.cgi) for the depiction of conserved motifs within IAA domains (accessed on 30 September 2023). For homology modeling of three representatives of phylogenetic clad proteins (SsIAA15c, SsIAA23, and SsIAA18a), sequences were submitted to SWISS-MODEL (https://swissmodel.expasy.org/) (accessed on 1 October 2023) in auto-modeling mode with the best templates selected by the webtool itself. The high-scoring models for each protein were visualized in Chimera 1.15 software [45].

Furthermore, the promoter region sequences of SsIAA genes, spanning 2000 bp upstream of the translational start site (ATG), were extracted from the *Saccharum spontaneum* L. genome. Prediction of cis elements within the promoters was achieved through the utilization of the PlantCARE tool (https://bioinformatics.psb.ugent.be/webtools/plantcare/html/) (accessed on 28 September 2023)), while the visual representation of these elements was generated using Tbtools v2.016.

### 4.5. Hierarchial Clustering of RNA-Seq Gene Expression

Sugarcane SsIAA protein sequences from the v20190103 genome were blasted in the Saccharum Genome Database (http://sugarcane.zhangjisenlab.cn/viroblast/viroblast.php, accessed on 10 October 2023). The resultant gene IDs corresponding to an earlier published *S. spon* genome [17] (AP85-441) were identified. The SsIAA gene IDs of AP85-441 were used to find the RNA-seq expression in three datasets (http://sugarcane.zhangjisenlab.cn/sgd/html/mRNA.html, accessed on 21 October 2023) based on fragments per kilobase of exon per million fragments mapped (FPKM) with three biological replicates. The mean FPKM values from the aforementioned database were used to draw heat maps and hierarchical clustering in TB tools.

### 4.6. RNA Extraction and qPCR

Sugarcane leaves were accurately weighed (0.1 g) and fully ground in liquid nitrogen, following the strict operating instructions of the RNAprep Pure Plant Plus Kit (Polysaccharides Polyphenolics-rich, DP441, TIANGEN, Beijing, China). The purity and concentration of RNA were determined using a microspectrophotometer. Qualified RNA was used for cDNA synthesis, which was performed strictly according to the instructions for using the RNA PCR Kit v2.1 (TaKaRa, Dalian, China), employing AMV Reverse Transcriptase XL as the enzyme. Finally, the concentration and purity of cDNA were determined using a trace spectrophotometer.

For stress studies, 3-week-old seedlings were grown in a compost–vermiculite mixture from culms of wild-type *S. Spon*. (grown in the fields of Guangxi University) containing single buds. After 3 weeks, the seedlings were shifted into trays containing double distilled water and acclimatized for one week. For cold stress treatments, seedling trays were placed in 6 °C growth chambers, and samples were collected at 0, 1, 3, 6, and 12 h intervals. For salt stress, a 200 mM NaCl solution was applied, and samples were collected at the same intervals as cold stress. Initial growth of seedlings and salt stress were conducted in a greenhouse at 28 °C, with 16 h of light and 8 h of darkness. At each treatment interval, 8–10 seedlings were used for sample collection. QRT-QPCR was conducted with the ABI 7500 Detection System (Applied Biosystems, Foster City, CA, USA) using a SYBR GREEN PCR Master Kit (TaKaRa). Primer 5.0 software was used to design quantitative real-time PCR primers for genes. The specific primers are listed in Appendix A. The *SsGADPH* gene was used as an internal control. The 2^−ΔΔCT^ method was utilized to calculate mean expression levels and standard deviation (SD).

### 4.7. Construction of IAA-GFP Vectors and Subcellular Localization Observations

Cloning primers were designed using the Vazyme primer designing tool (https://crm.vazyme.com/cetool/en-us/singlefragment.html, accessed on 10 March 2024) for single-digest KpnI restriction sites for the pCAMBIA1300-35S-GFP vector using *SsIAA23* and *SsIAA12.1a* coding sequences without stop codons. The primers are listed in Appendix A. The coding sequences of *SsIAA23* and *SsIAA12.1a* genes without stop codons were PCR-amplified with high-fidelity P525 polymerase using GT42 cDNA as a template. Meanwhile, pCAMBIA1300-35S-GFP vectors were linearized with KpnI (Takara, Shiga, Japan). The linearized vector and *SsIAA23* and *SsIAA12.1a* PCR products were ligated using the Uniclone One Step Seamless Cloning Kit (SC612). The cloned vectors were introduced into competent cells of the DH5α strain of *E. coli* through the heat-shock method. Following the identification of positive colonies through colony PCR, sequencing was performed to confirm the presence and orientation of CDS. The plasmid extraction was performed using an Invitrogen Plasmid extraction kit. The empty and cloned vectors were introduced into Agrobacterium tumefaciens strain GV3101 and co-transformed with a chromatin marker (H2B::mCherry) into *Nicotiana benthamiana* leaves. The transformed plants were kept in darkness for 60 hour periods. The leaves were taken after 60 h and using a confocal laser scanning microscope (FV12000MPE, Olympus, Tokyo, Japan) the fluorescence signal was observed as mCherry emitted light at 600–650 nm and was excited at 552 nm, while GFP emitted light at 500–530 nm and was excited at 488 nm [46].

## Figures and Tables

**Figure 1 ijms-25-07473-f001:**
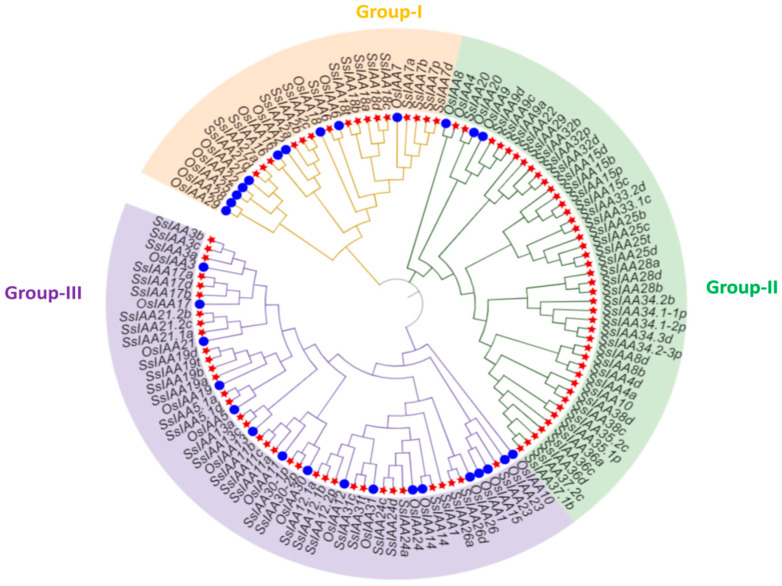
The phylogenetic relationships among IAA proteins of sugarcane and rice. The sugarcane proteins are designated by red stars, while blue circles represents rice IAA proteins.

**Figure 2 ijms-25-07473-f002:**
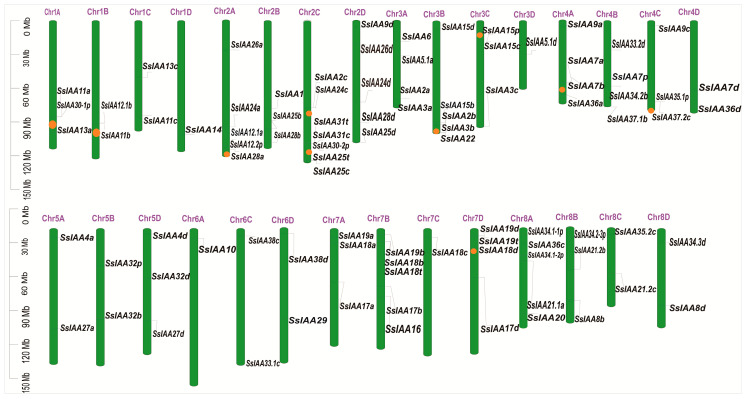
Distribution of the IAA gene family members on *Saccharum spontaneum* chromosomes.

**Figure 3 ijms-25-07473-f003:**
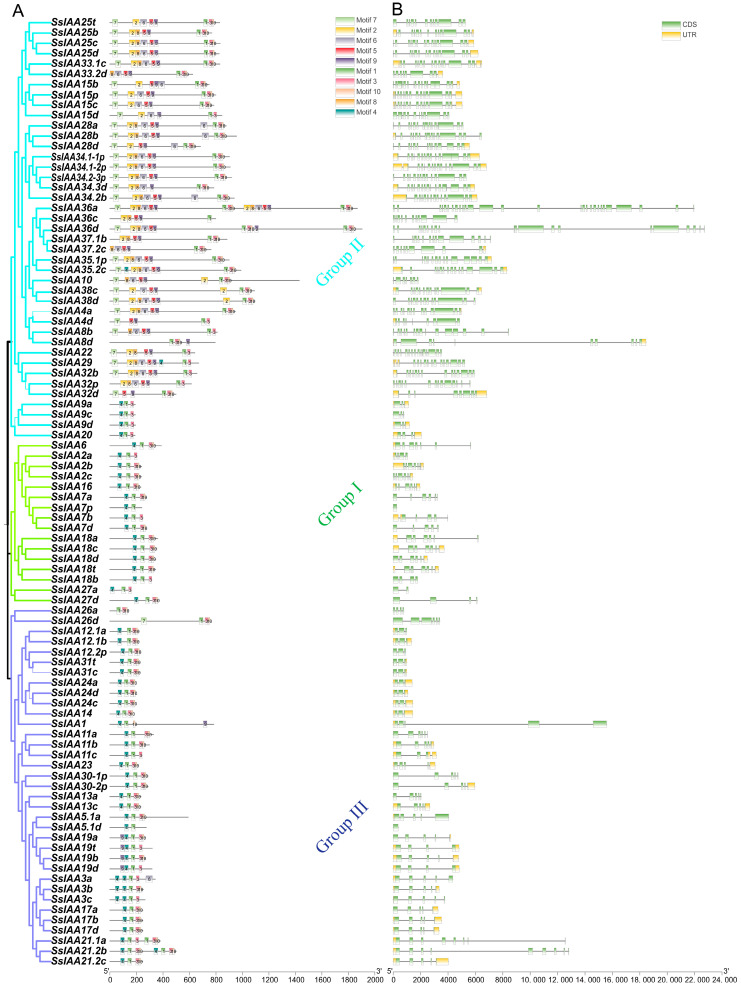
Conserved proteins motif composition (**A**) and gene-structure organization (**B**) of the *SsIAA* gene family. Ten conserved protein motifs are exhibited by different colors, CDS depicted by green, UTR by yellow, whereas intros are represented by straight lines.

**Figure 4 ijms-25-07473-f004:**
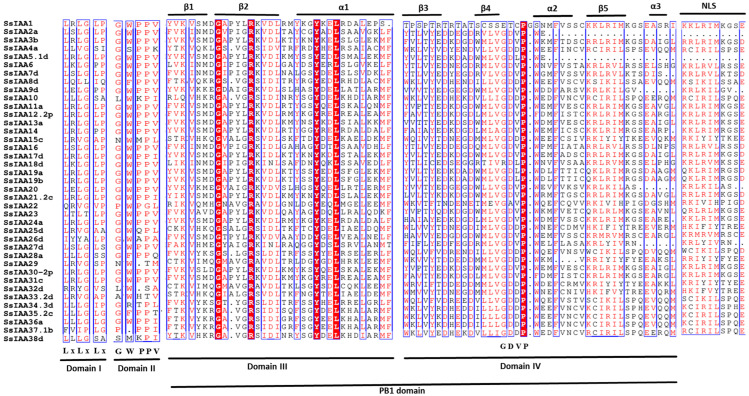
Multiple-sequence alignment (MSA) of SsIAA sequences. The conserved domains (I, II, III, and IV) of the SsIAA family are underlined. Text below each indicates signature conserved amino acid motifs. The conserved secondary structures’ beta chains and alpha helixes in domains III and IV are indicated with β and α, respectively. Red lines indicate completely conserved aminoacids, whereas blue lines depicts partially conserved residues.

**Figure 5 ijms-25-07473-f005:**
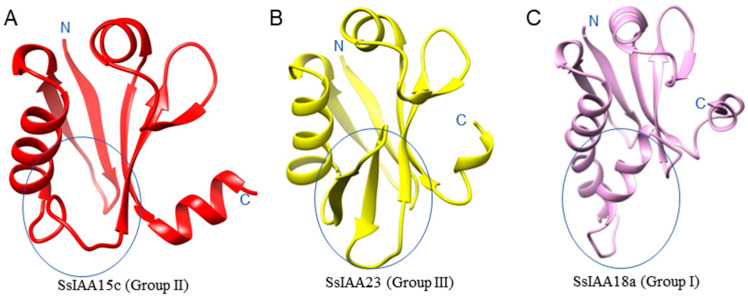
Tertiary structure analysis of IAA domains of three phylogenetic groups. The structural regions exhibiting variations are marked with ovals. (**A**) Phylogenetic group II is represented by SsIAA15c (**B**) Group III is represented by SsIAA23, and (**C**) Group I representation through SsIAA18a proteins.

**Figure 6 ijms-25-07473-f006:**
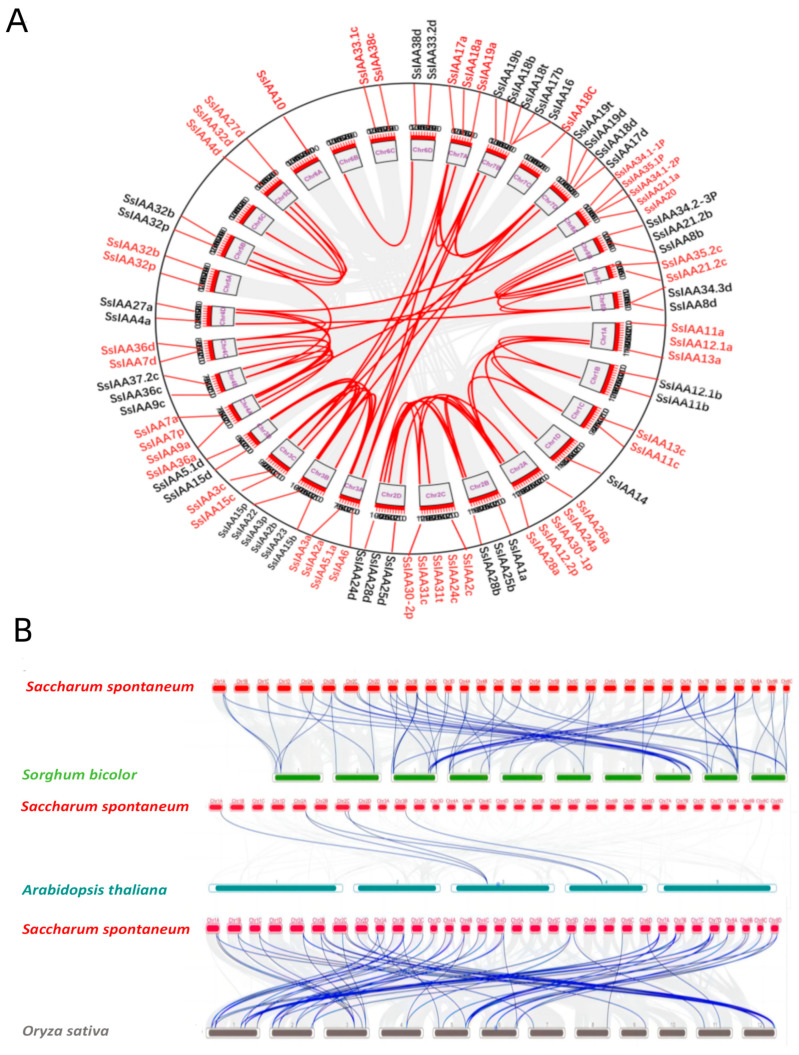
Collinear relationships of the sugarcane *IAAs* within its genome and with sorghum, *Arabidopsis*, and *Oryza sativa*. (**A**) Intraspecific collinear analysis of the sugarcane *IAA* gene family. The red lines indicate the collinear relationship of *IAA* genes, and the gray lines indicate the collinear relationship of all genes. (**B**) Collinearity analysis of sugarcane *IAA* genes between Sorghum bicolor, *Arabidopsis thaliana,* and *Oryza sativa*. The blue lines indicate the collinear relationship of *IAA* genes between the two species, and the gray lines indicate the collinear relationship of all genes among the two species.

**Figure 7 ijms-25-07473-f007:**
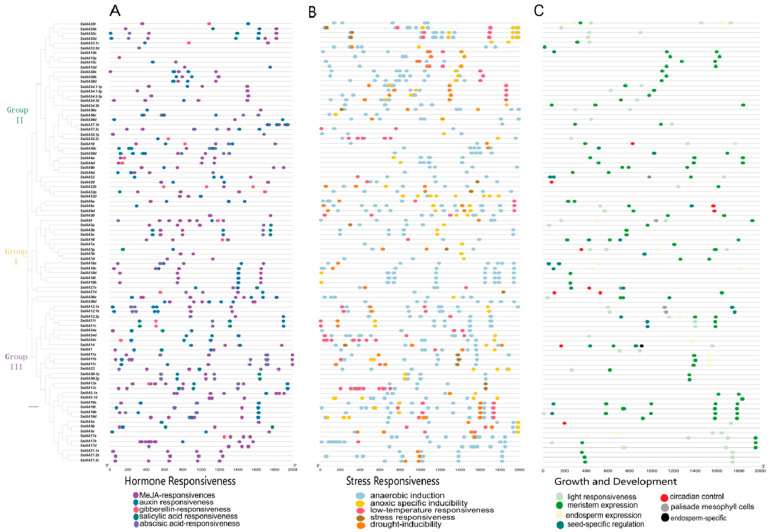
Cis-regulatory element (CRE) distribution on the predicted promoter regions of SsIAAs. (**A**) Hormonal responsiveness. (**B**) Stress responsiveness. (**C**) Growth- and development-related.

**Figure 8 ijms-25-07473-f008:**
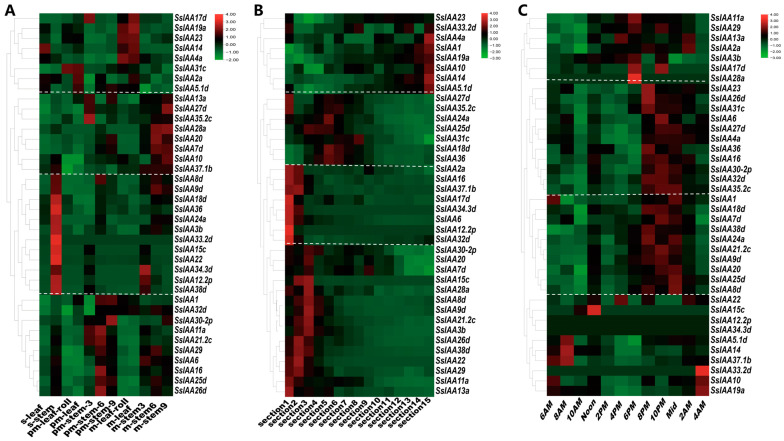
Hierarchical clustering of 38 *SsIAAs’* spatio-temporal expression dynamics based on FPKM (fragments per kilobase of transcript per million fragments mapped). (**A**) Tissues at different developmental stages. (**B**) Leaf developmental gradient. (**C**) Circadian rhythms. Abbreviations: s, seedling stage; pm, pre-mature stage; m, mature stage; r leaf, roll leaf; m leaf, mature leaf. The numbers 1–15 represent 15 leaf sections, each corresponding to 1 cm in length. Day–night circadian rhythm showed 12 time points with the interval of two hours. The scale on the right side of each heatmap displays the gene expression levels; red indicates higher, green depicts lower, and black exhibits medium levels.

**Figure 9 ijms-25-07473-f009:**
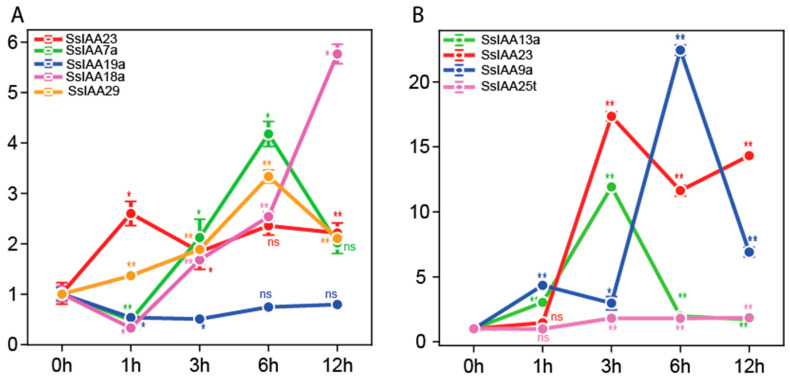
*SsIAA* gene expression pattern analysis utilizing qRT-PCR during 0, 1, 3, 6, and 12 h of salt- (**A**) and cold- (**B**) stress treatments. When presenting data, the mean ± standard deviation is used. The *t*-test indicates significant differences (* *p* < 0.05 and ** *p* < 0.01), ns = non significant.

**Figure 10 ijms-25-07473-f010:**
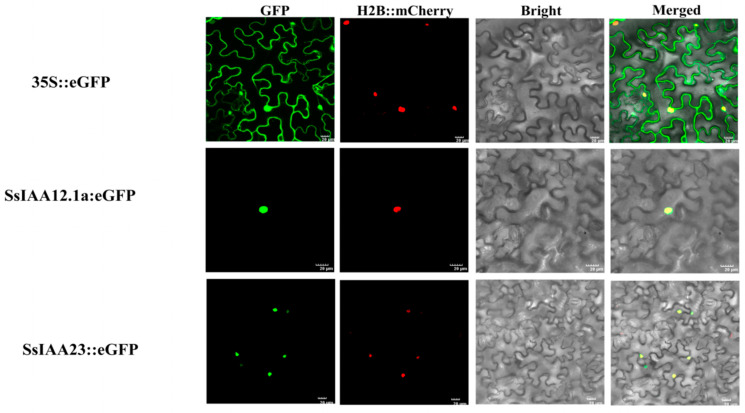
Subcellular localization of the SsIAA23 and SsIAA12.1a proteins in tobacco leaves expressing green fluorescent protein (GFP). H2B::mCherry was used as a chromatin marker. Bars = 20 μm.

**Figure 11 ijms-25-07473-f011:**
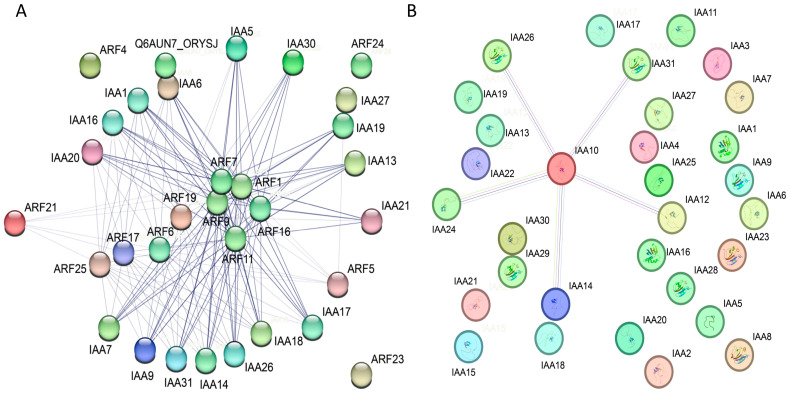
The predicted protein–protein interaction (PPI) networks of SsIAAs based on rice orthologs. (**A**) SsIAAs PPIs with ARFs proteins, and (**B**) SsIAAs PPI among themselves.

**Table 1 ijms-25-07473-t001:** Basic information of IAA gene family in sugarcane (*Saccharum spontaneum*).

Gene Name	Gene ID	Chromosome	Number of Amino Acids (aa)	Molecular Weight (kD)	Isoelectric Point	Instability Index	Subcellular Localization	Gene Duplication	Physical Position on the Genome	Strand Orientation	Biological Process (GO Term)
*SsIAA1*	Sspon.02G0041760-1B	Chr2B	783	82.65	9.44	64.91	Nucleus	Dispersed	77885470-77900915	Negative	AUX/IAA domain
*SsIAA2a*	Sspon.03G0022760-1A	Chr3A	204	21.05	5.38	48.17	Nucleus	Dispersed	69521520-69522442	Positive	Regulation of transcription, DNA-templated
*SsIAA3a*	Sspon.03G0024820-1A	Chr3A	342	36.48	9.34	44.68	Nucleus	Dispersed	75376437-75380644	Positive	Response to auxin
*SsIAA4a*	Sspon.05G0001870-1A	Chr5A	947	104.19	5.87	61.16	Nucleus	Dispersed	5939018-5943933	Positive	Root development
*SsIAA5.1a*	Sspon.03G0011430-1A	Chr3A	589	62.87	10.52	52.91	Nucleus	Dispersed	31092210-31096179	Positive	Regulation of transcription, DNA-templated
*SsIAA6*	Sspon.03G0006790-1A	Chr3A	388	42.06	9.15	44.89	Nucleus	WGD or segmental	18637083-18642587	Negative	Response to heat
*SsIAA7a*	Sspon.04G0014810-1A	Chr4A	278	29.59	7.8	49.33	Nucleus	Tandem	55340442-55343662	Positive	Response to auxin
*SsIAA8d*	Sspon.08G0022500-3D	Chr8D	793	85.86	5.71	54.34	Chloroplast	Unknown	49305894-49314316	Negative	Response to heat
*SsIAA9a*	Sspon.04G0001230-1A	Chr4A	191	20.44	5.49	42.22	Nucleus	WGD or segmental	4697296-4698075	Negative	Regulation of transcription, DNA-templated
*SsIAA10*	Sspon.06G0002150-1T	Chr6A	1429	155.9	5.77	57.27	Nucleus	Singleton	6930821-6932644	Negative	Response to heat
*SsIAA11a*	Sspon.01G0023790-1A	Chr1A	330	34.98	5.29	64.09	Nucleus	WGD or segmental	85454695-85457182	Positive	Regulation of transcription, DNA-templated
*SsIAA12.1a*	Sspon.01G0023800-1A	Chr1A	217	23.1	8.7	48.53	Nucleus	Singleton	85497372-85498224	Positive	Regulation of transcription, DNA-templated
*SsIAA13a*	Sspon.01G0028160-1A	Chr1A	234	25.18	8.86	48.73	Nucleus	WGD or segmental	90932535-90933387	Negative	Regulation of transcription, DNA-templated
*SsIAA14*	Sspon.02G0041760-3D	Chr1D	175	18.8	6.73	35.71	Nucleus	Dispersed	101374456-101375168	Positive	Regulation of transcription, DNA-templated
*SsIAA15b*	Sspon.03G0027750-1B	Chr3B	748	83.41	7.03	56.43	Nucleus	WGD or segmental	6658010-6662331	Negative	Regulation of transcription, DNA-templated
*SsIAA16*	Sspon.03G0022760-1P	Chr7B	233	23.99	6.76	42.19	Nucleus	Dispersed	45460339-45461913	Negative	Regulation of transcription, DNA-templated
*SsIAA17a*	Sspon.07G0010810-1A	Chr7A	248	26.71	7.68	41.01	Nucleus	Unknown	35935896-35938803	Positive family	Regulation of transcription, DNA-templated
*SsIAA18a*	Sspon.07G0003900-1A	Chr7A	359	38.42	5.84	52.94	Nucleus	Unknown	9728691-9733926	Positive	Regulation of transcription, DNA-templated
*SsIAA19a*	Sspon.07G0002380-1A	Chr7A	267	27.95	6.2	47.95	Nucleus	Unknown	5913783-5917916	Positive	Regulation of transcription, DNA-templated
*SsIAA20*	Sspon.08G0013920-1A	Chr8A	189	20.54	5.99	60	Nucleus	Unknown	57584849-57586099	Positive	Regulation of transcription, DNA-templated
*SsIAA21.1a*	Sspon.08G0007190-1A	Chr8A	379	42.07	8.99	38.47	Nucleus	Unknown	22454558-22466916	Positive	Response to heat
*SsIAA22*	Sspon.03G0037540-1B	Chr3B	639	71.77	5.72	58.42	Nucleus	WGD or segmental	99443945-99447482	Positive	Regulation of transcription, DNA-templated
*SsIAA23*	Sspon.03G0036500-1B	Chr3B	214	23.15	7.77	48.78	Nucleus	WGD or segmental	90332265-90334872	Negative	Regulation of transcription, DNA-templated
*SsIAA24a*	Sspon.02G0026740-1A	Chr2A	191	20.4	6.31	38.52	Nucleus	Dispersed	94923088-94923850	Positive	Regulation of transcription, DNA-templated
*SsIAA25b*	Sspon.02G0027120-2B	Chr2B	768	85.17	5.97	64.77	Nucleus	WGD or segmental	95779585-95785095	Negative	Regulation of transcription, DNA-templated
*SsIAA26a*	Sspon.02G0009510-1A	Chr2A	139	15.24	4.62	32.78	Nucleus	WGD or segmental	26751431-26752199	Negative	Regulation of transcription, DNA-templated
*SsIAA27a*	Sspon.05G0016400-1A	Chr5A	162	17.22	5.87	34.82	Nucleus	Singleton	66902764-66903857	Positive	Regulation of transcription, DNA-templated
*SsIAA28a*	Sspon.02G0031340-1A	Chr2A	878	96.97	5.94	64.59	Nucleus	WGD or segmental	114616279-114621376	Positive	Regulation of transcription, DNA-templated
*SsIAA29*	Sspon.04G0010200-2B	Chr4B	668	74.42	5.89	61.13	Nucleus	WGD or segmental	26207214-26212018	Negative	Negative regulation of transcription, DNA-templated
*SsIAA30-1p*	Sspon.01G0023790-1P	Chr2A	288	30.28	5.18	57.26	Nucleus	WGD or segmental	111385647-111390381	Positive	Regulation of transcription, DNA-templated
*SsIAA31c*	Sspon.01G0023800-3C	Chr2C	226	24.03	8.24	54.97	Nucleus	Dispersed	108456291-108457177	Negative	Regulation of transcription, DNA-templated
*SsIAA32b*	Sspon.05G0028050-1B	Chr5B	655	72.49	5.87	59.41	Nucleus	Dispersed	60122402-60128086	Positive	Regulation of transcription, DNA-templated
*SsIAA33.1c*	Sspon.06G0032700-1C	Chr6C	828	90.87	6.6	52.21	Nucleus	Proximal	91761546-91766954	Positive	Response to abscisic acid
*SsIAA34.1-1p*	Sspon.04G0017640-1P	Chr8A	901	99.99	5.93	68.72	Nucleus	Unknown	4675614-4680912	Negative	Response to heat
*SsIAA35.1p*	Sspon.04G0018530-1P	Chr8A	899	99.48	6.36	57.25	Nucleus	Unknown	6040168-6047060	Negative	Regulation of transcription, DNA-templated
*SsIAA36a*	Sspon.04G0018530-1A	Chr4A	1867	207.23	6.25	57.06	Nucleus	WGD or segmental	66589636-66611591	Negative	Regulation of transcription, DNA-templated
*SsIAA37.1b*	Sspon.04G0030300-1B	Chr4B	883	98.89	5.48	51.09	Nucleus	Dispersed	76690693-76697801	Negative	Regulation of transcription, DNA-templated
*SsIAA38c*	Sspon.06G0019860-2C	Chr6C	1091	120.82	6.14	57.95	Nucleus	Dispersed	5171377-5177422	Positive	Response to heat

Note: 38 groups of *SsIAA* were identified based on the orthologous rice genes, and each of the representative genes in the 38 groups is shown in this table. All 92 genes are shown in Appendix A. Singleton means that the gene is single-copy. Dispersed means that the gene might arise from transposition, such as “replicative transposition”, “non-replicative transposition”, or “conservative transposition”. Proximal means that the gene might arise from small-scale transposition or arise from tandem duplication and insertion of some other genes. WGD or segmental means that the gene might arise from whole-genome Duplication or segmental duplication. Unknown means did not find any record.

## Data Availability

Data is contained within the article and Appendix A.

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
