# Peer review of "Genome-Wide Analysis of the Auxin/Indoleacetic Acid (Aux/IAA) Gene Family in Autopolyploid Sugarcane (Saccharum spontaneum)"

_ijms, 2024, doi:10.3390/ijms25137473_

Round 1
Reviewer 1 Report
Comments and Suggestions for Authors
The manuscript presents a comprehensive analysis of the Aux/IAA gene family in autopolyploid sugarcane. The study is well-structured and provides a thorough examination of gene identification, phylogenetic relationships, gene structure, regulatory elements, expression profiles, and functional characterization based on transcriptome data. The inclusion of GFP-subcellular localization and qRT-PCR validation strengthens the findings. However, some areas could benefit from further clarification and improvement.
The introduction could be improved by explicitly stating the knowledge gap this study aims to fill regarding the Aux/IAA gene family in sugarcane. Additionally, the introduction would benefit from a clearer articulation of the study's objectives.
The methods section is detailed and provides a clear description of the procedures used. However, Please see the below comments for further improvement:
The criteria for identifying the Aux/IAA gene family members should be more explicitly stated. what thresholds were used for identification?
The source of the transcriptome data should be mentioned. Were biological replicates used?
More details on the GFP-tagging procedure and imaging techniques would be beneficial.
The results and discussion parts are well-organized and presented with appropriate use of figures and tables; following points could be taken into consideration for further improvements;
The discussion on the distribution of genes across chromosomes is informative but could be enhanced by comparing it to other species.
For the expression data the potential functional implications of differential expression in various tissues and conditions could be further elaborated.
qRT-PCR validation of results should include a detailed discussion on the consistency of these results with the transcriptomic data.
Few sentences on the potential applications of these findings in sugarcane breeding and stress resistance could be added.
Additional comments;
- Line 78: "Understanding how these proteins function can provide valuable insights for developing stress-tolerant crops." This statement would be more impactful if followed by specific examples of how this understanding has been applied in other crops.
- Table 1: The table lists the basic information of the IAA gene family members but does not provide functional annotations which could be helpful for readers.
Author Response
Comments 1; [The introduction could be improved by explicitly stating the knowledge gap this study aims to fill regarding the Aux/IAA gene family in sugarcane. Additionally, the introduction would benefit from a clearer articulation of the study's objectives.]
Response 1: [Thank you for pointing this out. We agree with this comment. Therefore, we have stated the research gap and objectives. The modifications can be found at page 2, paragraph 3, and lines 84 to 97].
Comment 2: [The criteria for identifying the Aux/IAA gene family members should be more explicitly stated. what thresholds were used for identification?]
Response 2: [Agree. We have written the thresholds used for the identification of SsIAA proteins. The modifications can be found on page 19, paragraph 3, and lines 473-474.]
Comment 3: [The source of the transcriptome data should be mentioned. Were biological replicates used?]
Response 3: [ We agree with the reviewer's comments. The source of transcriptome datasets and replicate information can be found on page 20, paragraph 3, and lines 532-535.]
Comments 4: [More details on the GFP-tagging procedure and imaging techniques would be beneficial.]
Response 4: [We agree with this comment. More details about GFP tagging and imaging procedure have been elaborated on page 21, paragraph 2, and lines 564-572.]
Comment 5: [The discussion on the distribution of genes across chromosomes is informative but could be enhanced by comparing it to other species.]
Response 5: [We agree with this comment. The required changes have been made through a comparison of gene distribution with other species. The desired modifications can be found on page 17, paragraph 2, and lines 360-365.]
Comment 6: [For the expression data the potential functional implications of differential expression in various tissues and conditions could be further elaborated.]
Response 6: [We agree with this comment. Further elaborations and potential functional implications of differential expressions have been made. The modifications can be found on page 17, paragraph 5, Lines 399-405, and page 18, paragraph 1, and lines 410-418.]
Comment 7: [qRT-PCR validation of results should include a detailed discussion on the consistency of these results with the transcriptomic data.]
Response 7: [In this study, the purpose of our qRT-PCR here is not to verify the results of transcriptomic data. Since there is no data for abiotic stresses available online, so we do qPCR analysis to extend the knowledge of the expression profile of the IAA gene family under abiotic stresses. Furthermore, we have removed the term validation of gene expression.]
Comment 8: [Few sentences on the potential applications of these findings in sugarcane breeding and stress resistance could be added.]
Response 8: [We agree with this comment. The potential application of these findings in sugarcane breeding and stress resistance has been added. The modifications can be found on pages 18-19, paragraphs 4-1, and Lines 453-463.]
Additional comments 1: [Line 78: "Understanding how these proteins function can provide valuable insights for developing stress-tolerant crops." This statement would be more impactful if followed by specific examples of how this understanding has been applied in other crops.]
Additional comments response 1; [ We agree with this comment. Line 78 has been repositioned to lines 70-72 followed by specific examples of how this understanding has been applied in other crops.]
Additional comments 2: [Table 1: The table lists the basic information of the IAA gene family members but does not provide functional annotations which could be helpful for readers.]
Additional comments response 2: [Agree. Table 1 has been modified as directed by the reviewer. The modification can be found on page 3, line 110-111.]
Reviewer 2 Report
Comments and Suggestions for Authors
The work presented by Huang et al.,entitled “Genome-wide analysis of the auxin/ indoleacetic acid 2 (Aux/IAA) gene family in autopolyploid sugarcane 3 (Saccharum spontaneum)” certainly gives a new insight on IAA and its role associated with abiotic stresses. This article can be accepted after minor improvements.
1. Why cold and salt stresses only were taken to study the abiotic stresses.
2. The discussion part needs to be more explained with recent references.
3. Authors are suggested to explain a bit more about methods, especially stress study.
4. Authors are suggested to check all the references (especially formats) in list and text.
Author Response
- Comment 1: [Why cold and salt stresses only were taken to study the abiotic stresses.]
Response 1: [Thank you for pointing this out. We used salt and cold stress because these are two common stresses faced by sugarcane in the local climes of Guangxi province China. We especially, think that the cold is a major restriction to expanding the sugarcane planting area. So, we prefer to do these two in this study, however, we think your suggestion is valuable, and we will do more abiotic stresses later.]
- Comment 2: [The discussion part needs to be more explained with recent references.]
Response 2: [We agree with these comments. The discussion parts are more elaborated as suggested. The modifications can be found on “page 17-18, paragraph 2-5, lines 360-365, 399-405”, “page 18, paragraph 1, 4, lines 411-418, 429-430, 442-443, 453-463”]
Comment 3: [Authors are suggested to explain a bit more about methods, especially stress study.]
Response 3: [We agree with this comment. The methods have been explained as suggested. The detailed modifications of methods can be found on page 20, paragraph 3, Lines 532-535, 5476-554, and page 21, paragraph 1 and lines 564-572 have been added.]
Comment 4: [Authors are suggested to check all the references (especially formats) in the list and text.]
Response 4: [Agree. The references in the list and text are according to Journal format.]